# pH-Sensitive Silver-Containing Carbon Dots Based on Folic Acid

**DOI:** 10.3390/ma15051880

**Published:** 2022-03-03

**Authors:** Qinhai Xu, Kang Li, Peng Wang

**Affiliations:** Department of Chemistry, Renmin University of China, Beijing 100872, China; 2017000460@ruc.edu.cn (Q.X.); 2020102318@ruc.edu.cn (K.L.)

**Keywords:** Ag-containing carbon dots, pH sensor, fluorescence emission, structural mechanism, spectroscopy

## Abstract

Herein, Ag-containing carbon dots (Ag-CDs) was synthesized based on folic acid. In a neutral solution, its fluorescence emission owns a structure fixing fluorescent species with the emission maximum at 400 nm and an excitation-wavelength dependent fluorescent species, respectively. By comparing fluorescent emission and excitation spectra, the electronic absorption origins of these fluorescent species were assigned. With the assistance of UV–Vis absorption and XPS, the pH-regulating fluorescence mechanism of Ag-CDs was studied and proposed. A particularly strong fluorescence emitter was observed at pH ~12 with a mixing coordination structure as Ag(CDs-NH_2_)OH. The as-prepared Ag-CDs might be developed into a fluorescent sensor, especially at extremely basic conditions.

## 1. Introduction

Carbon dots (CDs), a kind of fluorescent carbon-based materials, have attracted great attention recently for a wide variety of chemical and biological applications, such as biosensing, bioimaging, drug delivery, photodynamic therapy, photocatalysis and electrocatalysis, due to their good fluorescence quantum yield, excellent solubility, high biocompatibility, nontoxicity, and good photochemical stability [1,2,3,4]. Generally, almost all CDs show excitation-dependent fluorescence [5]. Among many proposed origins of the excitation-dependent fluorescence, the surface functional group of CDs draws the most attention. Nie et al. suggested that the surface functional groups of CDs, such as C=O and C=N, can efficiently introduce new energy levels for electron transitions and contribute to the main origin of excitation-dependent fluorescence [6]. Wen et al. proposed that the excitation-wavelength-dependent fluorescence arises from the abundant carboxyl functional groups on the surface [7]. Wang et al. found that no matter what kind of method is used to fabricate CDs, all their fluorescence originates from special edge states consisting of several carbon atoms on the edge of the carbon backbone and surface carbonyl or carboxyl groups as well [8]. Lei et al. demonstrated that the functional groups with different degrees of oxidation on the surface of CDs lead to the variation in emission wavelength by changing the energy band gap of CDs [9]. Sharma et al. proposed that surface functional groups determining aggregation are the origin of discrete multiple electronic states for the excitation-dependent emission in carbon nanodots [10]. In addition to the excitation-dependent fluorescence, CDs sometimes have another shorter wavelength emissive component without excitation-dependent behavior. This excitation-independent shorter wavelength emission is found to depend on the states of surface amino-groups, while the excitation-dependent longer wavelength emission is due to the presence of carboxyl or hydroxyl groups [5,11]. The roles of N-doping in forming the emission states of CDs were reviewed by Wang et al. [2]. In this review, either N-doping directly into the carbon skeleton, or Mg/N, P/N, and S/N co-doping were demonstrated to be good strategies to bring more excellent properties to the resulting CDs. Some other researches showed that N-doping CDs could be developed into a potential fluorescent probe for detecting environmental Hg^2+^ ions [12] or pathogenic bacteria at an acidic pH [13]. Recently, Khan et al. reported N-doping green-emissive CDs synthesized via a solid-state reaction method with good thermal stability and much better quantum yield in ethanol than in water [14].

The fluorescence of CDs is usually found to be pH-dependent, which is attributed to the reversible protonation and deprotonation of functional groups on the surface of CDs [15,16,17,18,19] and the aggregation of CDs [17,20,21]. Among these, N-doping also plays a very important role. For example, CDs prepared by Maillard reaction with L-tryptophan as N resources own excellent photoluminescence stability and stable pH-dependence and could be used to selectively detect Cr (VI) in tap water [22]. Jute-derived fluorescent surface-quaternized CDs were developed as fluorescent nanobuttons to detect aqueous chromium (VI) and a pH-responsive drug release carrier [23]. CDs with polyethyleneimine (PEI) as a surface passivation agent exhibits pH-responsive optical properties [24]. 

In addition, the environmental metal ions affect the fluorescence of CDs too, and this has been used to develop a CDs-based metal ion sensor. Among the metal elements studied, Ag^+^ was found to significantly enhance the fluorescence of CDs by chelating Ag^+^ with surface functional groups [25,26,27]. Therefore, directly doping Ag^+^ into CDs during preparation is a promising way to develop Ag-containing CDs (Ag-CDs) [28,29,30].

The doped Ag^+^ may occupy part of the surface functional groups of CDs, which generally are also sensitive to the environmental pH values. However, the pH effect on the fluorescence behavior of Ag-CDs has not been studied yet. In this study, a hydrothermal method was applied to synthetize Ag-containing carbon dots (Ag-CDs) by using folic acid (FA) as a carbon source and sliver nitrate as a silver source. The compositional and structural characterizations by Fourier transform infrared absorption spectroscopy (FT-IR) and X-ray photoelectron spectroscopy (XPS) confirmed the primary amine and carboxyl groups co-existing on the surface of prepared Ag-CDs. By comparing their UV–Vis absorption, fluorescence emission and excitation spectra, the electronic absorption origins of the two fluorescent species observed in neutral solution were assigned. With the assistance of UV–Vis absorption and XPS, the pH effect on the fluorescence of Ag-CDs was studied. A very strong fluorescence species observed at pH ~12 was deduced to be with mixing coordination structure as Ag(CDs-NH_2_)OH. Finally, the pH-regulating fluorescence mechanism of Ag-CDs was proposed. It is worth noting that two of the three fluorescent species are Ag-containing. 

In this research, Ag-containing CDs based on folic acid were prepared, and a more detailed mechanism of its pH-regulating fluorescence was studied and proposed. This contribution might be developed into a fluorescent sensor, especially in extremely basic conditions. 

## 2. Experimental Section

### 2.1. Reagents and Apparatus 

All the needed reagents and apparatus in the studies were shown in Appendix A.

### 2.2. Preparation of Ag-Containing Carbon Dots (Ag-CDs) 

Ag-doped carbon dots (Ag-CDs) were prepared by the one-step hydrothermal method. Firstly, 0.1 g of folic acid (FA) and 0.2 g of AgNO_3_ were dissolved in 30 mL of ultra-pure water. Then, the solutions were transferred into an autoclave (50 mL) and heated at 200 ℃ for 12 h. The resulting suspensions were centrifuged at 15000 rpm for 10 minutes, filtered by micro-filtration membranes (ϕ = 0.22 μm) three times, and then dialyzed (MWCO 1000 Da) for one day. The purified Ag-CDs were obtained and stored. 

### 2.3. Effect of pH on the Fluorescence of Ag-CDs 

To investigate the effect of pH on the fluorescence properties of Ag-CDs, concentrated NaOH and H_2_SO_4_ aqueous solutions were used to adjust the pH values of Ag-CDs solution as 0.74, 1.32, 2.25, 3.09, 4.06, 5.06, 5.89, 6.37, 7.00, 8.19, 9.04, 9.99, 11.10, 12.15, 12.97, and 13.50, respectively. The pH value was measured with a digital pH meter (PHS-3C, YOKE INSTRUMENT, Shanghai, China). The reversibility of the effect of pH on the fluorescence of Ag-CDs was investigated by changing the pH values between pH = 12.15 and pH = 2.25 by repeatedly adding concentrated NaOH or H_2_SO_4_. The fluorescence emission spectra were recorded by the fluorescence spectrophotometer (HITACHI F-4600, HITACHI, Tokyo, Japan).

### 2.4. pH Fluorescent Test Paper

In order to evaluate the practical application ability of Ag-CDs as a pH fluorescent probe, the filter paper was soaked in Ag-CDs solutions for 1 minute and then dried at 90 °C for 2 h to obtain the pH fluorescent test paper. To determine the feasibility of the test paper in various water samples, tap water and ultrapure water were chosen for comparison. Three pH values were selected for the tests, i.e., 11.15, 12.20 and 13.02. The prepared test papers were soaked in various water samples for 2 s and were taken out immediately. After staying in the air for 10 min, the fluorescent photographs were taken under UV light.

## 3. Results and discussion

### 3.1. Morphology and Composition of Ag-CDs

To study the morphology of Ag-CDs prepared in this research, a TEM image was measured, as shown in Figure 1a. The TEM image (Figure 1a) of Ag-CDs shows that the prepared Ag-CDs own good dispersity and the particle size had an average diameter of 3.23 ± 1 nm (insert (I) in Figure 1a). In addition, the lattice spacing of crystallized Ag-CDs was determined as 0.21 nm corresponding to the (110) diffraction plane of graphitic carbon (JCPDS cards 26-1076), as shown as insert (II-1) in Figure 1a [31].

To obtain the functional group information, the FT-IR spectrum of Ag-CDs was measured and is presented in Figure 1b. The moderate and broad bands peaked at 3186 and 2834 cm^−1^ were assigned as the stretching modes of O-H in the carboxyl group and C-H on sp^3^ carbons, respectively, and the broadening and down-shifting of the former peak were due to the hydrogen bonds in existing intra- or inter-particles [32,33]. The strong band peak at 1682 cm^−1^ was attributed to the stretching mode of the C=O group in the carboxyl group [34]. The peaks at 1574 and 1516 cm^−1^ were assigned as the C=C stretching modes of the graphitic backbone of Ag-CDs [35]. The peaks at 1337 cm^−1^ could be ascribed to the stretching modes of C-N, which originated from the primary and secondary amine groups attached to the graphitic backbone [36]. The peaks at 1184, 1095, and 1041 cm^−1^ are assigned as the stretching modes of C-O [6,17,35].

The elemental composition of Ag-CDs was determined by XPS (the full XPS spectrum of Ag-CDs is shown in Appendix A), and the results for C1s, Ag3d, N1s, and O1s are shown in Figure 1c–f, respectively. In C1s’ XPS spectrum (Figure 1c), three peaks at 284.8, 286.2, and 288.4 eV were assigned as the carbon atoms in C-C, C-O, and O-C=O groups, respectively [12,33,37,38,39]. In the Ag3d spectrum (Figure 1d), the peaks at 367.7 and 373.78 eV were attributed to Ag3d_5/2_ and Ag3d_3/2_, respectively, which indicated the presence of Ag^+^ [26,40]. In N1s’ spectrum (Figure 1e), the peak at 400.0 was assigned as the N atoms in -NH-/-NH_2_ groups [32,38]. In O1s’ spectrum (Figure 1f), the peaks at 531.7 and 532.6 eV were attributed to the carbon atoms in C=O and C-O groups, respectively [33]. In combination with all the results of FT-IR and XPS, primary and secondary amino (-NH_2_/-NH-), carboxylic (-COOH), and hydroxyl (-OH) groups were determined to attach to the graphitic carbon backbone of Ag-CDs prepared in this research. 

### 3.2. Assignment of the Electronic Absorption Origins of the Fluorescent Species of Ag-CDs in Neutral Solution 

Figure 2a shows the absorption spectrum of Ag-CDs in a neutralized aqueous solution. A moderate absorption band range in wavelengths from 275 to 350 nm and a weaker shoulder range from 350 to 500 nm were observed. The normalized fluorescence emission spectra of Ag-CDs under excitation at various wavelengths (λ_EX_) are shown in Figure 2b. Upon excitation at wavelengths of 270–330 nm (thinner black line in Figure 2b), the spectrum seems almost the same as the single band peaked at ~400 nm, indicating that a structure-fixing luminescent species exists in this spectral region. When λ_EX_ is ≥ 350 nm and moves toward red, the emission spectra show characteristic excitation-wavelength-dependent spectral behavior, i.e., moving towards red too. This typical fluorescent species observed in many carbon dots was mainly attributed to the excitation-wavelength selecting size dependence of emissive traps on the surface of CDs [41,42]. Notably, upon excitation at ~350 nm, the emission spectrum shows dual-emission characteristics. Such spectral pattern was used as a standard region for pH effect study in the following.

The electronic absorption origins of fluorescent emission of Ag-CDs were studied by fluorescent excitation spectra. The normalized spectra (Figure 2c) were categorized into two groups based on fluorescent probing wavelengths (λ_pr_). For the first group, i.e., λ_pr_ in 350–400 nm, the normalized excitation spectra have two bands. The one in the shorter wavelength range (250–330 nm) that peaked at 294 nm has almost the same spectral pattern on the blue edges, which mainly corresponds to the structure-fixing fluorescent species and is further ascertained as in 270–330 nm, as shown in Figure 2b. Another one is just constantly red-shifting and intensity-enhancing upon λ_pr_ red-shifting. For the second group, i.e., λ_pr_ in 460–550 nm, the normalized fluorescence excitation spectra have almost the same spectral pattern in the blue region (250–330 nm) and varied ones in the red region (330–500 nm), especially the red-edges, which gradually shift toward longer wavelengths. This fluorescence species and the second one in the first group were assigned together, similar to the typical excitation-wavelength-dependent luminescent species aforementioned.

To summarize, two fluorescent species were determined to exist in Ag-CDs in neutral solution. One has a fixed electronic structure with absorption in 250-330 nm, and the other is an excitation-wavelength-dependent species with absorption in 250-500 nm. 

### 3.3. pH-Regulating Fluorescent Variation in Ag-CDs

The pH effect on fluorescent properties of Ag-CDs was studied by measuring their UV–Vis absorption and fluorescence emission spectra at various pH values in a range of 0.74–13.50. The results are summarized in Figure 3. Under daylight (Figure 3a-I), the aqueous solutions of Ag-CDs are almost colorless at pH < 12.15 but gradually become yellowish when pH ≥ 12.15. Correspondingly, under UV light 365 nm (Figure 3a-II), all solutions exhibit blue fluorescence. At extremely acidic and alkaline conditions, i.e., pH < 2.25 and >12.15, the fluorescent emission becomes obviously weaker, while at pH 12.15, the utmost bright fluorescence is observed, and at all the other pH values, the fluorescent emission seems the same as each other. 

The fluorescence spectra of Ag-CDs under λ_EX_ 360 nm at various pH values were shown in Figure 3b, which were consistent with the luminescent photos in Figure 3a-II. At extremely acidic conditions (pH < 2.25), the spectra show a single-peak band with the emission maximum (λ_max_) at 400 nm; at pH in 2.25–11.10, the fluorescent spectra show uniform dual-emission characteristics with peaks at 400 and 450 nm, respectively; at pH ≥ 12.15, the fluorescent spectra have a single-peak band with λ_max_ at 460 nm, and the fluorescent intensity significantly enhances at pH 12.15 and decreases quickly upon the pH value increasing further. It is worth pointing out that the fluorescent emission at 450 nm observed in pH 2.25–11.10 has a different origin from the emission at 460 nm observed in pH ≥ 12.15. A more detailed discussion is in Section 3.4. 

Fluorescent intensities, probed, respectively, at 400 and 450 nm were depicted in Figure 3c versus the variation of pH values. Apparently, the 450 nm is a more sensitive probing wavelength for studying pH-effect on fluorescence. Along with the pH change from 0.74 to 13.50, three stages were clearly seen, i.e., pH at 0.74–2.25, 2.25–11.10, and 12.15–13.50, respectively. The fluorescent intensity correspondingly undergoes the following changes, i.e., gradually increasing, keeping constant, and significantly enhancing and dropping down soon.

The absorption spectra of Ag-CDs at various pH were shown as in Figure 3d. At a pH between 0.74 and 11.10, the spectra seem unchanged. At three extremely alkaline conditions, a different spectral pattern was observed. The obvious collapse of the 300–350 nm peak and the significant increase in the 350–500 nm shoulder (Figure 3d) indicate that a structural change happens particularly upon the pH changing from 11 to 12.

### 3.4. Structural Mechanism of pH-Regulating Fluorescence of Ag-CDs

The results of FT-IR and XPS (Section 3.1.) indicate that primary amine (-NH_2_), -COOH, -OH, and Ag^+^ coexist in the Ag-CDs prepared in this research. By spectral analysis in a neutral solution, as mentioned in Section 3.2., two fluorescent species were deduced to coexist in a neutral solution of Ag-CDs. One has a fixed structure with electronic absorption in the range of 250-330 nm, and the other has wavelength-dependent fluorescent emission properties with absorption in the range of 250–500 nm. The further pH-regulating fluorescence study (Section 3.3.) proves that the extremely acidic and alkaline conditions will cause structural changes, which either quench the fluorescence species of 450 nm or generate a new fluorescence species emitting at 460 nm. Combining all the aforementioned results, we proposed a pH-regulating and Ag^+^-containing fluorescent mechanism, as shown in Figure 1. 

At a pH in 2.20–11.15, two fluorescent species coexist in Ag-CDs, i.e., the one free of Ag^+^ with characteristic fluorescence at 400 nm and the other with the Ag^+^(-NH_2_)_2_ group and characteristic fluorescence at 450 nm, which were named FL-1 and FL-2, respectively. These two fluorescent species contribute to the typical dual-emission spectra measured in a neutral solution. When the pH is at an extremely acidic condition, i.e., pH ≤ 1.32, the amino group of FL-2 is thoroughly protonated, and the originally coordinated Ag^+^ might be dissolved in the extremely acidic solution. This causes the fluorescence emission to stop at 450 nm (seeing Figure 3b,c). When the pH is at 12.15, the concentrated OH^–^ may compete with –NH_2_ in coordinating with Ag^+^ to form a new intermediate, as proposed in Figure 1 as FL-3. The significantly enhancing fluorescence intensity and characteristic emission at 460 nm differentiate FL-3 from FL-2. Furthermore, at this pH value, the fluorescence of FL-1 and its corresponding absorption vanish together, indicating that an extremely alkaline condition causes the structural change and the consequential fluorescence quenching of FL-1. When the pH increases further, the fluorescence intensity of Ag-CDs drops quickly, but the spectral pattern remains unchanged because a more alkaline condition will cause the formed AgOH to quickly transform into Ag_2_O irreversibly, which is supported by the yellowish color of solutions at extremely alkaline conditions seen under daylight (Figure 3a).

High-resolution X-ray photoelectron spectroscopy (XPS) of Ag3d and N1s at various pH values (Figure 4) were used to support the proposed mechanisms above. 

As shown in Figure 4a, at a pH of 1.32, no Ag^+^ signal was observed. This could be explained by the thorough protonation of the amino group by the extremely acidic conditions (as shown as FL-1 in Figure 1), which causes Ag^+^ to dissolve into the solutions. Upon the pH increasing, the gradual deprotonation of –NH_3_^+^ results in the formation of Ag(-NH_2_)_2_^+^ again, while FL-1 and FL-2 coexist and generate a dual-emission spectral at pH 2.25–11.10. At a pH of 2.25, which is the ending point of the transition stage from aqueous Ag^+^ to Ag(-NH_2_)_2_^+^, the Ag^+^(-NH_2_)(H_2_O) might exist too. Since the filling capability of a lone pair from an oxygen atom of H_2_O is apparently weaker than that from a nitrogen atom of –NH_2_ groups, there might be smaller electron binding energy (BE) of Ag3d at pH 2.25 (the red line in Figure 4a) than that at neutral pH (the blue line in Figure 4a). When pH arrives at 12.15, a new intermediate state forms as proposed as FL-3 in Figure 1. Higher BE of Ag3d is observed (purple line in Figure 4a). This could be explained by one coordination bond between Ag^+^ and -NH_2_ being replaced by a covalent bond Ag-O. When basicity increases further, AgOH may transform into Ag_2_O irreversibly, which decreases the fluorescence intensity of FL-3 and exhibits a yellowish color correspondingly (seeing the photographs at daylight at pH ≥ 12.15 in Figure 3a-I). Reasonably, the XPS signal of Ag3d disappears again at pH 12.97 since most of the coordinated Ag^+^ has transformed into Ag_2_O. The XPS of N1s of Ag-CDs at the same pH values is shown in Figure 4b. At a pH of 1.72, the amino group mainly exists in –NH_3_^+^ form, and the positive charge causes the BE of N1s to be the highest one (black line in Figure 4b). When the pH is at 2.25 (red line in Figure 4b), the transition from –NH_3_^+^ to coordination with Ag^+^ results in a free –NH_2_-like state. This is very close to the extremely alkaline state (pH 12.97, brown line in Figure 4b) since all the -NH_2_ becomes free again due to the almost-complete precipitation of Ag_2_O. When the pH is neutral or at 11.10 (blue and green lines in Figure 4b), the coordination state as Ag(-NH_2_)_2_ has middle BE values between free -NH_2_ and –NH_3_^+^ states since the positive charge of Ag^+^ has a weaker binding capability to electrons than that of H^+^ but a larger one than a free –NH_2_ state. When the pH is at 12.15, the amine group is at the transition state of Ag(-NH_2_)OH, and the electron donation effect of a negative charge of ^–^OH weakens the bound capability of Ag^+^ to electron compared with that at Ag(-NH_2_)_2_ state. That is why the BE value at pH 12.15 (purple lines in Figure 4b) is smaller than that at a neutral state.

### 3.5. Fluorescent Reversibility Ag-CDs upon the Variation in pH

The very bright fluorescence emission of Ag-CDs at pH 12.15 and the sensitive pH effect on fluorescence probed at 450 nm, as shown above, drove us to explore the potential of Ag-CDs used as a fluorescent probe at extremely alkaline pH values. Figure 5 presents the repeatability and reversibility test of fluorescent intensity changing between pH 2.25 and 12.15. The as-prepared Ag-CDs are subjected to pH cycling between pH = 12.15 and pH = 2.25 using NaOH and H_2_SO_4_ solutions repeatedly. As shown in Figure 5, the fluorescence intensities decrease significantly upon a pH change from 12.15 to 2.25. Afterwards, the FL intensity was restored nearly as before when the pH was changed back from 2.25 to 12.15. The luminescence switching operation could be repeated for four consecutive cycles without fatigue, indicating the good reversibility of the pH fluorescent sensor based on Ag-CDs. The photo-stability of Ag-CDs was evaluated by continuously measuring the fluorescence intensity probed at 400 and 450 nm, respectively, under excitation at 360 nm. The results are shown in Appendix A, which indicated that Ag-CDs prepared in this research have good photo-stability [16,29].

This result implies that Ag-CDs in this research have good reversibility upon variation in pH values, which facilitates it to be developed as a fluorescent sensor practically at a very broad pH range.

### 3.6. pH Fluorescent Test Paper

As displayed in Figure 6a, the filter paper was soaked in Ag-CDs solutions and dried to obtain the pH fluorescent test paper, and the fluorescent photograph was taken upon excitation at 365 nm. The prepared pH fluorescent test papers were then soaked in water samples, including blank and the ultrapure water samples at various pH values (11.15, 12.20, and 13.02), and were taken out immediately. The photographs were taken under excitation at 365 nm. From Figure 6b(a-3), compared with other test papers, the pH fluorescent test paper soaked in ultrapure water at pH = 12.20 showed the brightest fluorescence, indicating that the pH fluorescent test paper could be a sensitive pH-sensing device under strongly alkaline conditions. After 10 min under daylight, the pH fluorescent test paper was measured again, and the results shown in Figure 6b(a-4) indicated very good photo-stability of the prepared fluorescent sensor based on Ag-CDs. In addition, the pH fluorescent test paper was evaluated in a real sample, i.e., tap water, too. As shown in Figure 6b(b-3), the results for sensing pH variation in tap water were consistent with those in ultrapure water. This result proves that the pH fluorescent test paper based on Ag-CDs prepared here could be developed into a pH sensor with practical applications under strongly alkaline conditions.

## 4. Conclusions

The Ag^+^-doped CDs were synthesized through the hydrothermal method in this work. The structural mechanism has been studied through various spectroscopic methods under a broad range of pH values. The primary amine, carboxyl groups, and Ag^+^ were determined to coexist in the as-prepared Ag-CDs by FT-IR and XPS. The electronic absorption origins of the two fluorescent species in the neutral solution have been assigned through a comparison between fluorescent emission and excitation spectra. With the assistance of XPS at various pH values, the pH-regulating fluorescence mechanism was proposed. The most interesting finding is a very bright fluorescent species observed at pH 12.15, and the further fluorescent reversibility test upon pH variation and the pH fluorescent test paper prove that the Ag-CDs prepared in this study might be developed into a fluorescent sensor, especially at extremely basic conditions. This work also paves the road for developing a similar pH-regulated metal ion-doped CDs-based fluorescent sensor.

## Data Availability

The data will be made available on request.

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
