# Peer review of "pH-Sensitive Silver-Containing Carbon Dots Based on Folic Acid"

_materials, 2022, doi:10.3390/ma15051880_

Round 1

Reviewer 1 Report

The paper presents a detailed report on the pH behavior of silver-doped carbon dots obtained from folic acid. An interesting pH dependence of fluorescence is presented and discussed in connection with excitation, emission and XPS spectra. A feasibility of using the presented behavior in sensing is emphasized.

The following questions and remarks appear when reading this article.

  1. It sounds complicated and unclear. What does it mean “Ag centered”? It is sufficient to call it “Ag-containing”. Why “dual emission”? It is shown in line 235 and Scheme 1 that there are THREE emitters. Why “carbon QUANTUM dots”? They do not have the properties of quantum dots (independence of the emission wavelength on the excitation wavelength etc) – we call them just “carbon dots”, please see the titles of the reviews: in 2014 they are “carbon QUANTUM dots”, whereas in 2021 the reviews are entitled “carbon dots”. Overall, I would suggest to change the title for the following: “pH-sensitive silver-containing carbon dots based on folic acid”
  2. The role of nitrogen in forming the emissive states of carbon dots is not shown. Only oxygen-containing groups are considered.
  3. Why only pH 12.15 was studied? What happens if pH is 12.05, 11.9, 12.25…? Fig. 3c looks strange with one point standing out at pH 12.15.
  4. Line 231 – silver sulfate never precipitates from an aqueous solution unless the concentration is much higher that you used. The mentioning of this precipitate must be removed from Scheme 1.
  5. Again, dual emissive or triple emissive? Please be consistent in Scheme 1 and in text.
  6. Language should be polished.

The paper can be published after revision.

Author Response

Dear Editors:

We do appreciate all the reviewers and editors for their comments and contributions. We have revised the manuscript with ID: materials-1581802. All the revisions have been highlighted with yellow color except some trivial revisions concerning spelling and grammars. We also prepared a text file named as “comments and responses” as “point by point” form.

We do thank for your kindly dealing with this revision at your earliest convenience.   

Best regards.

Yours sincerely,

Peng Wang

Name: Peng Wang (Associate Professor)

E-mail: wpeng_chem@ruc.edu.cn

Reviewer 2 Report

The authors clearly present the idea of creation a simple fluorescence sensor.

To this aim, they have synthesized Ag doped carbon quantum dots and firstly explored in details their structural properties. Further, the fluorescence properties were explored along with the effect of pH on optical properties.

Finally, a sensor is produced and its efficiency and repeatability was demonstrated.

This MS is clearly written and presented and therefore I recommend it for publication.

1. A brief summary (one short paragraph) outlining the aim of the paper and its main contributions.

This paper reports on the study of effect of pH Regulating Fluorescence Mechanism of dual-emission Ag- Centered Carbon Quantum Dots. The title describes the study. The abstract is concise and reflects the manuscript. The overall structure of the manuscript is good and written in a fluent language. The finding, in general, are good and capable of contributing to the literature.

2. Broad comments highlighting areas of strength and weakness. These comments should be specific enough for authors to be able to respond.

The authors clearly present the idea of creation a simple fluorescence sensor.

To this aim, they have synthesized Ag doped carbon quantum dots and firstly explored in details their structural properties. The procedure of CQDs formation are clearly presented and ready for a separate check. Further, the procedure of examination of pH effect on CQDs are presented in details as well as preparation of pH fluorescent test paper.

The morphology of QDs is explored with TEM, FTIR and XPS which gives the clear picture of QDs structure. The optical properties are studied with UV-Vis abs spectra and fluorescence along with the effect of pH on these optical properties.

Finally, a sensor is produced and its efficiency and repeatability was demonstrated.

In conclusion this MS describes in details all steps from the starting idea to the final application of possible detector and opens a way for possible similar pH-regulated metal ion doped CQDs based fluorescent sensor.

I do not see any weakness in this MS.

This MS is clearly written and presented and therefore I recommend it for publication.

3. Specific comments referring to line numbers, tables or figures. Reviewers need not comment on formatting issues that do not obscure the meaning of the paper, as these will be addressed by editors.

No comment.

Author Response

(The authors gave the same response as above.)

Reviewer 3 Report

In this manuscript, the author reports, ‘pH Regulating Fluorescence Mechanism of dual-emission Ag Centered Carbon Quantum Dots’. The current study is on a topic of relevance and general interest to readers in this area. The authors should address the following questions before getting a possible publication.

Recommendation: Major revisions needed as noted.

  1. The novelty of the present work should be discussed in the Introduction section.
  2. The inset of the Fig.1 is not visible clearly to the readers.
  3. The author should write purpose for each test in one/two sentences (in brief) before explaining the results of the characterization techniques. Therefore, the logic and organization of this part will be enhanced.
  4. The authors are encouraged to provide XPS full scan of the Ag-CQDs.
  5. What about photostability of the Ag-CQDs? What is the quantum yield of the Ag-CQDs?
  6. The formatting and grammatical errors in the article need to be checked carefully.
  7. The authors are encouraged to provide one comparison table discussing about the present work with the previously reported studies.
  8. The authors cited some of the relevant research works that have been conducted in this area however there are a few that needs to be included (shown below) in the Introduction section for better literature: Nanomaterials, 10(10), 1924 (https://doi.org/10.3390/nano10101924); ACS Applied Nano Materials, 3(12), 11777-11790 (https://doi.org/10.1021/acsanm.0c02305); ChemNanoMat, 1(2), 122-127 (https://doi.org/10.1002/cnma.201500009)

Author Response

(The authors gave the same response as above.)

Reviewer 4 Report

Dear Authors, 

This work describes pH Regulating Fluorescence Mechanism of dual-emission Ag-2 Centered Carbon Quantum Dots. This work shows interesting results. However, before I can recommend it for publication some corrections need to be addressed as follows.

Please fit the XPS data presented in the Fig 4 and assign all oxidation states/species present in the samples. Provide short discussion of the changes observed in the main manuscript.

Please check the grammar and spelling. All units should be in SI format.

Author Response

(The authors gave the same response as above.)

Round 2

Reviewer 3 Report

The authors have addressed all the questions raised before. Therefore the manuscript can be accepted in the present form

Reviewer 4 Report

Dear Authors,

The manuscript quality is more readable after corrections. After minor English spell check I can recommend this work for publication.

This manuscript is a resubmission of an earlier submission. The following is a list of the peer review reports and author responses from that submission.